# Oral Pathobiont-Derived Outer Membrane Vesicles in the Oral–Gut Axis

**DOI:** 10.3390/ijms252011141

**Published:** 2024-10-17

**Authors:** Eduardo A. Catalan, Emilio Seguel-Fuentes, Brandon Fuentes, Felipe Aranguiz-Varela, Daniela P. Castillo-Godoy, Elizabeth Rivera-Asin, Elisa Bocaz, Juan A. Fuentes, Denisse Bravo, Katina Schinnerling, Felipe Melo-Gonzalez

**Affiliations:** 1Laboratorio de Inmunología Traslacional, Facultad de Ciencias de la Vida, Universidad Andrés Bello, República 330, Santiago 8370186, Chile; ea.catalan.aguila@gmail.com (E.A.C.); emisefue@gmail.com (E.S.-F.); br.fuentesp@gmail.com (B.F.); f.aranguizvarela@gmail.com (F.A.-V.); daniel.castillo@uandresbello.edu (D.P.C.-G.); elizabethriveraasin@gmail.com (E.R.-A.); ebocazy@gmail.com (E.B.); 2Laboratorio de Genética y Patogénesis Bacteriana, Facultad de Ciencias de la Vida, Universidad Andrés Bello, República 330, Santiago 8370186, Chile; jfuentes@unab.cl; 3Cellular Interactions Laboratory, Faculty of Dentistry, Universidad Andrés Bello, Santiago 8370133, Chile; denisse.bravo@unab.cl

**Keywords:** pathobionts, OMVs, oral–gut axis, systemic disease, innate immunity, inflammation, dysbiosis, virulence factors

## Abstract

Oral pathobionts are essential in instigating local inflammation within the oral cavity and contribute to the pathogenesis of diseases in the gastrointestinal tract and other distant organs. Among the Gram-negative pathobionts, *Porphyromonas gingivalis* and *Fusobacterium nucleatum* emerge as critical drivers of periodontitis, exerting their influence not only locally but also as inducers of gut dysbiosis, intestinal disturbances, and systemic ailments. This dual impact is facilitated by their ectopic colonization of the intestinal mucosa and the subsequent mediation of distal systemic effects by releasing outer membrane vesicles (OMVs) into circulation. This review elucidates the principal components of oral pathobiont-derived OMVs implicated in disease pathogenesis within the oral–gut axis, detailing virulence factors that OMVs carry and their interactions with host epithelial and immune cells, both in vitro and in vivo. Additionally, we shed light on the less acknowledged interplay between oral pathobionts and the gut commensal *Akkermansia muciniphila*, which can directly impede oral pathobionts’ growth and modulate bacterial gene expression. Notably, OMVs derived from *A. muciniphila* emerge as promoters of anti-inflammatory effects within the gastrointestinal and distant tissues. Consequently, we explore the potential of *A. muciniphila*-derived OMVs to interact with oral pathobionts and prevent disease in the oral–gut axis.

## 1. Introduction

The oral cavity and the intestinal tract represent distinct microenvironments housing unique microbiomes [1]. While bacteria can traverse from the oral cavity to the intestinal tract through the ingestion of food and saliva, various physical and chemical barriers, including gastric acidity [2,3,4], bile acid secretion [5], and inherent resistance to colonization, serve to restrict the proliferation of oral bacteria in the gut. Nonetheless, the transmission of oral commensals to the gut appears to be more prevalent than previously acknowledged, as evidenced by the presence of oral microbes in the feces of healthy individuals, contributing to the modulation of the gut microbiome [6].

Conversely, the colonization of oral pathobionts in the gut can precipitate intestinal and systemic diseases. Prolonged use of proton pump inhibitors may induce alterations in the gut microbiome, concomitant with an elevated oral-to-gut transmission of oral microbes, such as *Fusobacterium nucleatum* (*Fn*), recognized as a pathobiont [7,8]. Similarly, the identification of oral bacteria, including *Fn*, in the gut has been linked to various conditions, such as colorectal cancer, inflammatory bowel disease (IBD) [9,10], and rheumatoid arthritis [11]. Moreover, ectopic colonization of oral *Klebsiella* strains in the gut triggers Th1-associated inflammation in gnotobiotic murine models [12], while a murine model of periodontitis induces gut dysbiosis through the colonization of salivary microbiota [13].

On the contrary, the periodontal pathobiont *Porphyromonas gingivalis* (*Pg*) exhibits minimal oral-to-gut transmission [6]. However, it has been associated with oral and gut dysbiosis in murine models of infection [14,15,16], provoking localized and systemic effects. Additionally, *Pg* may instigate an increase in the abundance of Bacteroidales, a reduction in the expression of tight junction proteins in the ileum, and subsequently elevated blood endotoxin levels. This phenomenon is correlated with heightened insulin resistance, translocation of *Enterobacteria* to the liver, and systemic inflammation [14,15]. *Pg*-induced dysbiosis is further characterized by diminished beneficial commensals, such as the mucin-degrading bacteria *Akkermansia muciniphila* in the cecum, potentially yielding systemic repercussions [16]. Notably, gut dysbiosis induced by *Pg* may contribute to systemic autoimmune diseases, such as rheumatoid arthritis, by driving pathogenic Th17 responses. This is evidenced by mice orally infected with *Pg* displaying increased susceptibility to experimental arthritis [17].

Gram-negative bacteria, such as *Pg* and *Fn*, release outer membrane vesicles (OMVs) that govern interactions within microbial communities and host cells at mucosal sites. In recent years, mounting evidence suggests that OMVs may have a crucial role in orchestrating the host immune response and facilitating the dissemination of bacterial components, including virulence factors, from mucosal sites to distant tissues. This process can potentially contribute to initiating and advancing systemic inflammatory diseases (Figure 1).

## 2. Composition and Function of Outer Membrane Vesicles (OMVs)

All Gram-negative bacteria can release spherical vesicles originating from the outer membrane, spanning a size range from 20 nm to 250 nm. Termed outer membrane vesicles (OMVs), these structures are notably enriched with outer membrane-associated proteins, lipopolysaccharides, and peptidoglycans. Moreover, they contain a spectrum of contents, encompassing periplasmic and cytosolic proteins, nucleic acids, as well as virulence factors such as toxins and enzymes (reviewed in [18]). While OMVs are continually secreted during the bacterial cell’s expansion phase, the specific composition of their contents is contingent upon environmental and culture conditions [19,20,21]. The proteomic analysis of OMV content presents inherent challenges, attributed not only to variations in bacterial growth conditions but also to factors related to OMV purification methods, gel separation, mass spectrometry strategies, and database search settings. This complexity is exemplified by comparing distinct approaches to characterize *Pg* OMVs, revealing a mere 50% overlap in identified proteins [22,23,24].

Oral pathobiont-derived OMVs have been shown to interact with a wide array of host membrane proteins found on both epithelial and immune cells. Most of the literature reviewed in this article highlights interactions with surface receptors that recognize pathogen-associated molecular patterns (PAMPs), which play a key role in modulating immune responses. In addition to receptor interactions, OMVs can be internalized by host cells through endocytosis, allowing the release of their cargo into the cytosol, where they may interact with cytosolic components and/or be trafficked to the lysosomal compartment. Moreover, some studies have demonstrated that OMVs can fuse directly with the host cell membrane, enabling bacterial proteins to remain associated with the host cell surface [25]. Despite these findings, the precise endocytic pathways involved in the recognition and uptake of OMVs derived from oral pathobionts remain poorly characterized. This review focuses specifically on the mechanisms described for OMVs from *Pg* and *Fn* OMVs.

OMVs serve diverse functions, including but not limited to cell-to-cell communication, survival under stress conditions, and the delivery of virulence factors (reviewed in [26]), which will be discussed below in the context of *Pg* and *Fn*.

## 3. *Porphyromonas gingivalis*

### 3.1. Porphyromonas gingivalis and Its Link to Periodontitis and Systemic Diseases

*Pg* is a strict anaerobic Gram-negative coccobacillus classified within the Bacteroidetes phylum. It assumes significance as a vital component of the oral microbiota and constitutes a constituent of the “red complex”, a group of periodontal pathogens. Functionally, *Pg* actively contributes to the proliferation of periodontal biofilm and holds recognition as an essential “keystone” pathobiont in periodontitis (PD), an inflammatory disorder characterized by the degradation of soft tissue and bone that supports the teeth [27]. Beyond its involvement in PD, the presence of *Pg* is linked to an extensive array of organ-specific and systemic diseases, including rheumatoid arthritis (RA), Alzheimer’s disease (AD), type 1 diabetes (T1D), cardiovascular disease (CVD), diabetic retinopathy [28], non-alcoholic fatty liver disease (NAFLD) [29], and various cancers. The latter encompass tumors of the digestive tract, esophageal cancer, colorectal cancer, pancreatic cancer, and hepatocellular carcinoma, as extensively reviewed [30].

### 3.2. Effects of Porphyromonas gingivalis in the Gut

Transmission of *Pg* to the gut is facilitated by its resistance to acidic pH, potentially fostering the development and progression of diseases by inducing microbial dysbiosis and disrupting gut epithelial integrity [31,32,33]. Oral administration of *Pg* heightened the severity of sodium dextran sulfate (DSS)-induced colitis in mice, disrupting the colonic epithelial barrier [32]. The capacity of *Pg* to diminish the expression of tight junction proteins, such as ZO-1, and enhance epithelial permeability was validated in Caco-2 cells in vitro and in the DSS-colitis model in vivo, contingent upon the activity of *Pg* gingipains [32]. A parallel compromise in barrier integrity—characterized by reduced ZO-1 and E-cadherin expression in the large intestine and endotoxin translocation to the bloodstream—was also noted in a mouse model of high-fat Western diet-induced non-alcoholic fatty liver disease (NAFLD) induced by oral infection with *Pg* [33]. In this diet-induced NAFLD model, *Pg* triggered persistent alterations in gut microbiota composition, marked by reduced community diversity and diminished abundance of short-chain fatty acid (SCFA) producers [31]. Additionally, a decrease in the phylum Firmicutes and the genus *Lactobacillus* and an increase in *Eubacterium fissicatena* were observed [33]. *Pg*-induced gut dysbiosis further exacerbated diet-induced hepatic lipid accumulation (steatosis) and glucose intolerance, thereby contributing to the progression of NAFLD [31,33]. Furthermore, *Pg* influenced the expression of genes in the liver, activating pathways associated with NF-κB signaling, endoplasmic reticulum stress, circadian rhythm, fibrosis, and tumorigenesis [33]. Consequently, *Pg*-induced gut dysbiosis may play a role in promoting the pathogenesis of distal liver diseases.

### 3.3. Porphyromonas gingivalis Outer Membrane Vesicles

*Pg* secretes OMVs of diverse sizes, spanning from 50 to 300 nm in diameter [34,35,36,37]. The composition of *Pg* OMVs encompasses lipopolysaccharide (LPS) and a delimited array of bacterial proteins derived from the cytoplasm, periplasm, and inner and outer membranes, constituting only 0.27% to 3.3% of the total *Pg* proteins [37,38]. The growth stage of the bacterium dictates the yield, protein composition, and presence of virulence factors on *Pg* OMVs [38]. Notably, OMVs obtained from pre-logarithmic cultures of *Pg* exhibit a greater morphological diversity and contain a higher abundance of proteins associated with bacterial metabolism, such as those involved in starch, sucrose, and glycerol phosphate metabolism. In contrast, OMVs collected from late-logarithmic and stationary phases are enriched with resistance-related proteins, including resistance to vancomycin, β-lactam, and cationic antimicrobial peptides [38]. Proteomic investigations have elucidated that *Pg* OMVs harbor various virulence factors, comprising proteolytic enzymes such as gingipains (Kgp, RgpA, and RgpB), *Pg* peptidyl arginine deiminase (PPAD), hemagglutinin (HagA), heme-binding protein (HBP35), and fimbrial adhesins (FimA, Mfa1) [24,39,40].

OMVs originating from *Pg* favor interactive communication with other oral pathobionts and can subsequently modulate the microbial community’s virulence. *Pg* OMVs facilitate horizontal gene transfer between *Pg* strains [41], impede biofilm formation [41], and engage in co-aggregation with other bacteria, including *Treponema denticola* and *Lachnoanaerobaculum saburreum* [42]. Additionally, *Pg* OMVs have the potential to interact with other periodontal pathobionts, thereby enhancing virulence and exacerbating the infection. These vesicles play a critical role in promoting the adhesion and invasion of the oral pathobiont *Tannerella forsythia* into host epithelial cells [43].

*Pg* OMVs are internalized by host epithelial cells via endocytosis in a Rac1/lipid raft-dependent mechanism, independent of dynamin, caveolin, and clathrin pathways [44]. Once internalized, OMV-containing early endosomes colocalize with lysosomal markers within 90 min of incubation yet evade degradation for 24 h while increasing the presence of acidified compartments [44]. Internalization of *Pg* OMVs via lipid rafts in non-phagocytic cells has been shown to activate pro-inflammatory responses through the host cytosolic receptor NOD1, which recognizes bacterial peptidoglycan. This activation leads to the stimulation of the NF-κB pro-inflammatory transcription factor, producing pro-inflammatory cytokines such as IL-8 [45].

Furthermore, *Pg* OMVs efficiently invade various host cell types [41], acting as carriers of virulence factors that disrupt normal cellular functions. One such study is *Pg* metallo-endopeptidase PepO, a homolog of human endothelin-converting enzyme (ECE)-1, which has been implicated in the invasion and intracellular survival of *Pg* in human gingival epithelial cells [46,47]. PepO is localized on the *Pg* membrane and has also been identified in *Pg* OMVs, suggesting its potential role in facilitating OMV entry into host cells [24,39,40].

The *Pg* OMVs elicit a diverse array of responses from pattern recognition receptors (PRRs) on host cells, leading to robust activation of membrane toll-like receptors 2 and 4 (TLR2 and TLR4), as well as moderate responses mediated by intracellular toll-like receptors 7, 8, and 9 (TLR7, TLR8, TLR9), and nucleotide-binding oligomerization domains 1 and 2 (NOD1 and NOD2). This has been confirmed through assays utilizing the respective HEK-Blue reporter cells [34]. Although multiple other host receptors can engage components present in OMV, various virulence factors in *Pg* OMVs contribute to the immune evasion mechanisms employed by *Pg* (Table 1), a topic that will be elaborated upon in the subsequent discussion.

#### 3.3.1. Gingipains

Gingipains are trypsin-like cysteine proteinases exhibiting arginine-specific activity (RgpA and RgpB) or lysine-specific activity (Kgp) (reviewed in [72]). These proteolytic enzymes are localized on the outer membrane and OMVs of diverse *Pg* strains, playing a crucial role in bacterial growth and survival through protein digestion for nutritional purposes. Additionally, gingipains contribute significantly to virulence functions. They facilitate adherence to epithelial cells, induce hemagglutination and hemolysis of erythrocytes, modulate the inflammatory response, and participate in the degradation of host proteins and tissue matrix, as extensively reviewed [72]. Notably, gingipains bound to OMVs exhibit higher stability than their soluble counterparts [73]. The systemic delivery of gingipains through OMVs is posited to potentially contribute to the promotion of inflammatory diseases.

Gingipains influence the inflammatory response within macrophages toward *Pg*, likely achieved through the degradation of secreted cytokines and membrane pattern recognition receptors [55,74]. In vitro experiments involving the inhibition of arginine- and lysine-specific gingipains in *Pg* (ATCC 33277 and W83) OMVs have demonstrated an augmentation in the secretion of IL-1β, IL-6, IL-8, MCP-1, and RANTES by U937-derived macrophages [55]. Exposure to *Pg* gingipains resulted in the diminished surface expression of CD14, a coreceptor of toll-like receptor 4 (TLR4), in murine RAW 246.7 macrophages. This reduction was correlated with decreased TNF-α secretion and phagocytic capacity upon challenge with live *Pg* [74]. Furthermore, gingipains associated with *Pg* OMVs have been identified as capable of inactivating and degrading crucial effector molecules released from neutrophil granules, including myeloperoxidase (MPO) and the cationic antimicrobial peptide LL-37. This activity impedes bacterial killing [51].

Proteolytically inactive RgpA has been shown to interact with membrane receptors such as epidermal growth factor receptor (EGFR), integrin α4β6 complex, and transferrin receptor (Tfr1), commonly organized within lipid rafts [75]. Recognition of inactive RgpA by EGFR activates the phosphatidylinositol 3-kinase (PI3K)-protein kinase B (AKT) signaling pathway, leading to the expression of pro-inflammatory cytokines IL-6, TNF-α, and IL-1β in monocyte-derived dendritic cells (moDCs) [75].

Remarkably, gingipains within *Pg* OMVs also impact the integrity of epithelial and endothelial barriers [56,76]. Surface-associated gingipains on *Pg* (W83) OMVs have demonstrated the ability to enhance endothelial permeability in monolayers of human microvascular endothelial cells (HMEC-1), presumably through the degradation of cell-cell adhesins such as PECAM-1 [56]. In support of this, a recent study uncovered a novel mechanism by which gingipain-containing *Pg* may induce dysfunction of the intestinal barrier. *Pg* OMVs were shown to be internalized by intestinal epithelial Caco-2 cells, where they release the protease gingipain Kpg into the cytosol, leading to the degradation of the tight junction protein occludin from the cytoplasmic side, ultimately increasing cellular permeability in vitro [57]. These findings suggest that gingipain-containing *Pg* OMVs can disrupt both endothelial and epithelial barrier integrity by degrading adhesins and tight junction proteins.

Gingipain-containing *Pg* OMVs can also contribute to systemic disease. In a zebrafish larvae model, systemic inoculation with OMVs from the *Pg* (W83) WT strain, but not from gingipain Kgp, RgpA, or RgpB-deficient mutants, resulted in the induction of systemic disease characterized by cardiac edema, enlarged yolk sack, and increased mortality. This observation underscores the central role of gingipains in mediating vascular leakage [56]. Moreover, it has been demonstrated that gingipains from *Pg* (ATCC33277), whether in a free form or associated with OMVs, elevate permeability in hematoencephalic cerebral microvascular endothelial cells (hCMEC/D3) by degrading tight junction proteins such as zonula occludens-1 (ZO-1) and occludin [76]. This discovery establishes a crucial connection between *Pg* and neurological disorders, as a permeable blood–brain barrier is known to promote cerebral inflammation and contribute to the development of Alzheimer’s disease.

The *Pg* gingipains Kgp, RgpA, and RgpB not only provide amino acids for bacterial metabolism but also have the potential to disrupt host amino acid synthesis in the liver. In murine models, *Pg* has been shown to colonize the liver, where transcription of these proteases increases. This elevation in gingipain activity can impair host amino acid metabolism, specifically reducing the biosynthesis of L-glutamate and L-glutamine [77]. Such metabolic alterations have been linked to the induction of ferroptosis, a form of programmed cell death contributing to the development of non-alcoholic fatty liver disease in mice following *Pg* oral administration [77]. Additionally, reports indicate that *Pg* gingipains interfere with insulin sensitivity and glucose metabolism in vitro in hepatic HepG2 cells by attenuating insulin-induced Akt/glycogen synthase kinase-3β (GSK-3β) signaling [36]. Thus, OMVs containing gingipains may play essential roles in the pathogenesis of systemic diseases in various distal tissues, including the brain and liver.

#### 3.3.2. Anionic Lipopolysaccharide (A-LPS)

*Pg* produces two distinct types of lipopolysaccharide (LPS) molecules: the more prevalent O-LPS and the anionic A-LPS (reviewed in [78]). Both variants comprise lipid A, anchoring LPS to the outer membrane and contributing to its toxicity, along with a core oligosaccharide. The primary distinction lies in the highly variable polysaccharide (PS) attached to the core. O-PS consists of repeating units of a single tetrasaccharide, whereas anionic polysaccharide (A-PS) is characterized by a phosphorylated branched mannan [79]. Additionally, lipid A from *Pg* A-LPS exhibits heterogeneity, presenting tetra-acylated forms of 1435 kDa and 1449 kDa (LPS_1435/1449_) and penta-acylated forms of 1690 kDa (LPS_1690_) [78]. These alterations in lipid A composition are suggested to be a *Pg* strategy to evade the host’s innate response in periodontal tissue, impeding PRR activation and subsequent pro-inflammatory cytokine production [80]. A-LPS has also demonstrated the ability to confer serum resistance by reducing susceptibility to complement-mediated killing [81]. Furthermore, A-LPS plays a central role in anchoring virulence factors, such as gingipains and PPAD, to the surface of OMVs [52]. Conversely, chronic exposure to *Pg*-LPS has been implicated in neuroinflammation in C57BL/6 mice, leading to cognitive impairment and an Alzheimer’s disease (AD)-like phenotype [82]. In this model, *Pg*-LPS activates the TLR4/NF-κB signaling pathway, inducing the expression of inflammatory cytokines TNF-α, IL-1β, IL-6, and IL-8 [83]. Therefore, A-LPS in *Pg*-derived OMVs can contribute to immune evasion by *Pg* and elicit pro-inflammatory effects in distal tissues.

#### 3.3.3. Fimbria

*Pg* fimbriae are filamentous appendages on the bacterial surface composed of fibrous proteins crucial for biofilm formation, auto-aggregation, co-aggregation with other oral bacteria, adhesion, and invasion into host cells (reviewed in [84,85]). Fimbriated *Pg* strains express both long fimbriae (major component FimA) and short fimbriae (major component Mfa1), both of which have been demonstrated to be abundant in *Pg* ATCC 33277-derived OMVs [39].

Studies have shown that the afimbrial *Pg* strain W83 and *fimA* mutants produce fewer OMVs than the fimbriated ATCC 33277 strain, suggesting a crucial role of FimA in the vesiculation process [39]. Notably, the invasive activity of OMVs largely depends on the minor components forming the tip of long fimbriae (FimC, FimD, and FimE) rather than on FimA [39]. FimA has also been demonstrated to activate TLR2 signaling in host cells, such as dendritic cells (DCs), resulting in autophagy-mediated killing of *Pg* [86]. Conversely, Mfa-1 targets DC-SIGN, a C-type lectin receptor that facilitates evasion from autophagy and lysosome fusion, promoting the survival and persistence of *Pg* within DCs [86].

#### 3.3.4. *Pg* Peptidylarginine Deiminase (PPAD)

Peptidylarginine deiminases (PAD) catalyze citrullination, the conversion of arginine residues in polypeptide chains to citrulline, accompanied by a loss of positive charge at this position [87]. In humans, five isoforms of PAD (PAD1, 2, 3, 4, 6) have been identified, each fulfilling distinct physiological and pathological roles. In bacteria, however, PPAD is only present in a select few species of the *Porphyromonas* genus [88]. Three forms of PPAD can be distinguished: a form bound to the outer membrane, a secreted soluble form, and a form attached to OMVs by A-LPS [52]. PPAD plays a role in various physiological processes in *Pg*, regulating proteolysis and arginine metabolism and contributing to the biogenesis of OMVs, biofilm formation, and surface translocation of *Pg* [53].

Despite the low sequence identity of approximately 30% between PPAD and human PAD, PPAD is proficient in citrullinating bacterial targets and native human proteins. This activity contributes to the pathogenesis of various diseases, including periodontitis, rheumatoid arthritis, and interstitial lung disease [89,90,91]. Unlike human PAD, whose activity is calcium dependent and directed towards internal arginine residues, PPAD preferentially citrullinates C-terminal arginine independent of calcium [92]. The gingipains RgpA and RgpB of *Pg* play a crucial role in the citrullination process, as they generate peptides with C-terminal arginine, serving as substrates for the posttranslational modification by PPAD [93].

A recent study on the citrullinome of *Pg* OMVs identified 78 citrullinated proteins, with 51 proteins possessing a signal peptide for translocation to the periplasm [22]. Several of the identified citrullinated proteins have been implicated in autoimmune processes in rheumatoid arthritis, the PGE2 synthesis pathway in gingival fibroblasts, inter- and intra-microbial interactions in biofilm, and the modulation of neutrophil and gingival epithelial cell responses [22]. Evidence suggests that PPAD citrullinates cell surface proteins, such as FimA, which function as TLR2 ligands [54]. Consistent with this, a significant association has been reported between the presence of specific antibodies against PPAD and the gingipain RgpA in the serum of patients with rheumatoid arthritis [94]. The study demonstrated a high positive predictive value for anti-PPAD (82.5%) and anti-RgpA (93.7%) antibodies in diagnosing rheumatoid arthritis [94]. As a result, these antibodies may serve as promising biomarkers for the identification and progression of rheumatoid arthritis.

Another study reported that PPAD on *Pg* OMVs interferes with the function of complement component C5 [50]. Treatment of C5 with *Pg* OMVs generated a citrullinated form of the anaphylatoxin C5a, while OMVs derived from a PPAD mutant only generated the native form of C5a. This suggests that gingipain Rgp cleaves C5 to C5a, while PPAD citrullinates the C-terminal arginine-74 residue of C5a [50]. Citrullinated C5a has been shown to exhibit a reduced chemotactic potential and impaired capacity to activate neutrophils and U937 monocytes expressing the C5a receptor [50].

#### 3.3.5. *Pg* Non-Coding RNAs

*Pg* OMVs have been found to contain microRNA-size small RNA (msRNA), which can exert epigenetic regulation in host cells, contributing to the pathogenesis of both periodontitis and systemic diseases. One such msRNA, sRNA45033, has been identified in *Pg* OMVs and shown to downregulate the expression of chromobox 5 (CBX5), a protein associated with heterochromatin, in human periodontal ligament cells. This alteration impacts host cell methylation, suppressing p53 methylation, thereby promoting apoptosis in periodontal cells. This process may contribute to periodontitis [58]. Additionally, other msRNA present in *Pg* OMVs have been shown in vitro to be delivered to Jurkat T cells, where they reduce the expression of cytokines such as IL-5, IL-13, and IL-15. This suggests that *Pg* OMVs may modulate host adaptive immune responses, further influencing disease outcomes.

### 3.4. Mechanisms of Host Cell Invasion

In comparison to *Pg* bacterial cells, *Pg*-derived OMVs exhibit a heightened capacity to invade various human cell types, including gingival epithelial cells (GEC), gingival fibroblasts (HGF), oral keratinocytes (HOK), umbilical vein endothelial cells (HUVEC), and immune cells [41]. *Pg* OMVs can be internalized into epithelial cells through actin-mediated or lipid raft-dependent mechanisms or by directly binding to host cells through activating PRRs [30,44,95]. Consistent with this, it has been demonstrated that the internalization of *Pg* OMVs is mediated by a Rac1-regulated pinocytic pathway involving GPI-AP1, Rac-1, the activation of PI3K, and actin polymerization. Importantly, this process is independent of caveolin, dynamin, and clathrin [44]. *Pg* OMVs that enter HeLa cells and immortalized human gingival epithelial (IHGE) cells localize in early endosomes and later in lysosomal compartments. This suggests that intracellular OMVs might undergo degradation by the cellular digestive machinery; however, they can remain within host cells for up to 24 h [44]. Remarkably, *Pg* OMVs have been shown to disrupt the function of cells upon internalization. Once inside, *Pg* OMVs induce the degradation of the transferrin receptor (TfR) as well as integrin-related signaling molecules such as paxillin and focal adhesion kinase. This disruption impacts iron homeostasis and impairs cell migration, critical processes for maintaining cellular function and integrity [96].

### 3.5. Effects of Porphyromonas gingivalis Outer Membrane Vesicles on Immune Cells

#### 3.5.1. Immunostimulatory Effects

OMVs derived from *Pg* have been demonstrated to elicit strong immunostimulatory effects by inducing the maturation of mouse bone-marrow-derived dendritic cells (BMDC). This, in turn, promotes IL-6-dependent differentiation of T cells toward a Th17 phenotype [35]. In murine RAW264.7 macrophages, *Pg* OMVs, particularly those recovered from late-log and stationary cultures, enhanced the expression of M1 macrophage biomarkers, including CD86 and inducible nitric oxide synthase (iNOS). Additionally, *Pg* OMVs triggered an increased transcription of genes encoding the inflammatory cytokines IL-1β, IL-6, and TNF-α in these cells [38]. A study on murine bone marrow-derived macrophages (BMDM) revealed that exposure to *Pg*-derived OMVs triggered a metabolic shift from oxidative phosphorylation (OXPHOS) to glycolysis, along with increased production of reactive oxygen species (ROS). Additionally, *Pg* OMV exposure led to the secretion of high levels of pro-inflammatory cytokines, including TNF-α, IL-12p70, IL-6, IL-10, IFN-β, and nitric oxide (NO). Inflammasome activation was evidenced as caspase-1 activation, lactate dehydrogenase (LDH) release, production of mature IL-1β and IL-18, and subsequent pyroptotic cell death [97]. In contrast, BMDM infected directly with *Pg* exhibited low levels of pro-inflammatory mediators and showed no signs of inflammasome activation or cell death [97]. These findings suggest that *Pg* OMVs are significantly more potent activators of innate immune responses than *Pg*.

#### 3.5.2. Immunoregulatory Effects

*Pg* OMVs may contribute to local immune evasion and the survival of *Pg* by inducing tolerance in monocytes and compromising neutrophil function [37,51]. It has been demonstrated that pre-stimulation with *Pg* OMVs induces the production of IL-10 in human peripheral blood monocytes, resulting in the suppression of TNF secretion upon re-stimulation with whole *Pg* bacteria [37]. This immunoregulatory effect of *Pg* OMVs on monocytes is dependent on surface TLR4 and mammalian target of rapamycin (mTOR) signaling and is independent of OMV endocytosis and the presence of fimbriae or gingipains [37].

In response to living bacteria, neutrophils typically release neutrophil extracellular traps (NETs), which capture and destroy extracellular bacteria through cytolytic enzymes. Interestingly, *Pg* (W83)-derived OMVs are not internalized by neutrophils and have been shown to selectively coat and activate human neutrophils, inducing degranulation without NET formation and bacterial killing [51]. Moreover, OMV-bound gingipains subsequently degrade secreted granule-derived antibacterial components such as LL-37 and myeloperoxidase (MPO), allowing bacterial survival [51].

The membrane lipids in OMVs may interact with host receptors; however, these interactions have yet to be investigated in the context of *Pg* and *Fn* OMVs. Sphingolipids within *Pg* OMVs can be recognized by host TLRs, promoting immunomodulatory properties in human macrophages [98]. Specifically, sphingolipids within *Pg* OMVs attenuate the pro-inflammatory response in THP-1 macrophage-like cells, influencing TLR2/MyD88-mediated secretion of IL-8, IL-6, IL-1β, and RANTES [98]. Host scavenger receptors, including the T cell immunoglobulin and mucin domain-containing 4 (TIM-4) receptor, have been shown to bind phosphatidylserine, a lipid typically exposed on the membrane of apoptotic cells, thereby enhancing the phagocytosis of exogenous bacterial particles by phagocytes in a phosphatidylserine-independent manner [99]. Notably, phosphatidylserine is not commonly found on the membrane of *Pg* or Fusobacteria [100]. Consequently, further research is warranted to explore the interactions between other lipids, such as phosphatidylethanolamine and phosphatidylglycerol, which may be present in OMVs derived from oral pathobionts and could mediate host–microbe interactions.

### 3.6. Effects of Porphyromonas gingivalis Outer Membrane Vesicles on Non-Immune Cells

*Pg* OMVs were demonstrated to induce IL-6 and IL-8 expression in the human gingival epithelial cell line OBA-9 through the Erk1/2, JNK, p38 MAPK, STING, and NF-κB signaling pathways [101]. Conversely, OMVs recovered from both late-log and stationery *Pg* (ATCC 33277) cultures exhibited potent inhibitory effects on proliferation, migration, and the expression of osteogenesis-associated proteins, such as alkaline phosphatase (ALP), Runx2, and Osterix (OSX), in human periodontal ligament stem cells (PDLSCs) [38].

In human retinal microvascular endothelial cells (HRMECs), stimulation with *Pg* OMVs increased the expression of inflammatory mediators TNF-α, IL-1β, IL-6, and MMP-9, ICAM-1, and the production of reactive oxygen species (ROS), induced mitochondria-related cell death, and altered endothelial permeability [28]. Most of these observed effects depended on the protease-activated receptor-2 (PAR-2) [28].

A recent study reported that *Pg* OMVs were internalized by first-trimester human trophoblast cells and reduced the oxidative stress response and expression of IL-8, IL-6, placental growth factor (PlGF), and vascular endothelial growth factor (VEGF-a), while increasing the expression of fms-like tyrosine kinase 1 (s-Flt-1) and the neutrophil and monocyte chemoattractant cytokine MCP-1 [102]. Conditioned media from *Pg* OMVs-treated trophoblast cells enhanced neutrophil chemoattraction and activation, indicated by the production of ROS, IL-8, and TNF, while reducing endothelial cell migration and increasing the adhesion of monocytes to endothelial cells [102].

### 3.7. Systemic Effects of Porphyromonas gingivalis OMVs

*Pg* OMVs can readily traverse barriers due to their small size and capacity to decrease the expression of intercellular adhesion molecules, facilitating their systemic dissemination. It has been demonstrated that *Pg* OMVs were detectable in the hippocampus and cortex of C57BL/6 mice three days after oral administration [49]. *Pg* OMVs diminished the expression of genes encoding tight junction proteins ZO-1, occludin, and claudin-5 in the hippocampus, inducing neuroinflammation [49]. This neuroinflammation was attributed to microglial activation via the NLRP3 inflammasome, leading to the secretion of the active form of IL-1β, subsequently promoting tau phosphorylation, neuronal degeneration, and cognitive impairment in mice [49]. Consistent with this, *Pg* OMVs have been shown to be internalized in microglia, leading to IL-1β via NF-κB activation in a reporter cell line. Interestingly, OMVs from *Pg* OMVs mutants lacking both gingipains Kgp and Rgp were neither phagocytosed nor capable of inducing IL-1β production, suggesting the OMV internalization is essential for this response [103]. However, pharmacological inhibition of both gingipains in WT *Pg* OMVs did not prevent IL-1β production upon stimulation [103], indicating that other components of *Pg* OMV likely contribute to this pro-inflammatory effect.

In a mouse model of diabetic retinopathy (DR) induced by intraperitoneal injection of streptozotocin into C57BL6/J mice, intravenous administration of *Pg* OMVs has been demonstrated to increase the permeability of the blood–retinal barrier and exacerbate the pathological microvasculature alterations [28]. A recent in vitro study also suggests that *Pg* OMVs impact trophoblast–endothelium and trophoblast–immune interactions, potentially contributing to pregnancy complications [102]. Both *Pg* and its OMVs have been shown to induce platelet aggregation, suggesting a potential contribution to the development of cardiovascular diseases, including atherosclerosis and myocardial infarction [104]. Therefore, *Pg* OMVs can exert pro-inflammatory effects due to their systemic dissemination, and further research is required to understand their effects on disease pathogenesis.

## 4. *Fusobacterium nucleatum*

### 4.1. Fusobacterium nucleatum and Its Link to Periodontitis and Systemic Diseases

*Fn* is an obligate anaerobic Gram-negative bacillus that resides in the oral cavity and has been identified as a contributor to periodontitis (reviewed in [105]). It serves as a pivotal component of the dental plaque biofilm, bridging primary colonizers such as *Streptococcus* spp. and secondary colonizers like *Pg* and *Aggregatibacter actinomycetemcomitans*, further exacerbating periodontitis (reviewed in [106]). Notably, amino acids derived from oral commensals *Streptococcus gordonii* and *Veilonella parvula* stimulate polyamine production by *Fn*. This, in turn, accelerates the *Pg* biofilm life cycle and dispersal, highlighting the intricate metabolic interactions between oral commensals and pathobionts that contribute to periodontitis [107].

As mentioned earlier, ectopic colonization of *Fn* in the gut has been linked to dysbiosis and colorectal cancer pathogenesis [9,10]. Oral inoculation with *Fn* does not induce intestinal inflammation in mice with normal microbiota. However, in mice treated with a broad-spectrum antibiotic cocktail, there is increased transcription of pro-inflammatory cytokines (IL-6, IFN-γ, and the murine homolog of IL-8, KC-1) at days 3 and 5 post-infection [108]. These findings suggest that the intestinal microbiota may counteract *Fn*’s pathogenic effects in the gut, but alterations in the microbiota may increase susceptibility to intestinal inflammation.

Moreover, other studies have demonstrated that *Fn* may play a role in the pathogenesis of IBD. Patients with both ulcerative colitis (UC) and Crohn’s disease exhibit an increased abundance of *Fn* in colon tissue, correlating with disease activity [109,110]. *Fn* upregulates the expression of caspase activation and recruitment domain 3 (CARD3), which can activate endoplasmic reticulum stress and promote gut epithelial barrier damage in a murine model of colitis [110]. Similarly, *Fn*-induced CARD3 upregulation can activate the pro-inflammatory NF-κB pathway through the upregulation of IL-17, contributing to the intestinal inflammation observed in IBD [109].

On the other hand, in addition to its role in colorectal cancer, recent studies have demonstrated that *Fn* may contribute to tumor metastasis to the liver. This is achieved through mechanisms involving the downregulation of intracellular tumor-suppressive microRNAs, the induction of colorectal cancer-associated alterations in the microbiota, and the modulation of liver immune responses [111,112]. Moreover, a recent investigation revealed the presence of *Fn* in liver samples from patients with acute liver failure (ALF), correlating with decreased energy metabolism and heightened liver inflammation [113]. In concordance with these findings, *Fn* infection in a murine model of ALF exacerbated liver inflammation by fostering macrophage infiltration into the liver [113]. These outcomes were attributed to *Fn*-mediated inhibition of the NAD+ synthesis pathway, ultimately promoting liver inflammation [113].

*Fn* can also induce host metabolic disruptions upon its translocation to the liver. In a mouse model of periodontitis, *Fn* not only exacerbated oral disease but also triggered hepatic alterations, notably in glycolysis and lipogenesis, via the PI3K/Akt/mTOR signaling pathway. These metabolic changes led to elevated systemic cholesterol and triglyceride levels, thereby enhancing atherogenesis and worsening atherosclerosis in the context of periodontitis [114]. In contrast, *Pg* did not induce lipogenesis or glycolysis in hepatocytes in vitro but promoted apoptosis [114]. These findings align with previous reports of disrupted hepatic amino acid biosynthesis during *Pg* colonization, suggesting that oral pathobionts can induce significant shifts in host metabolic processes.

In contrast to the well-established role of *Pg*-derived OMVs in systemic disease, much remains unknown about the involvement of *Fn*-derived OMVs in both the oral–gut axis and systemic disease. In this section, we will delineate recent findings in this field.

### 4.2. Fusobacterium nucleatum Outer Membrane Vesicles

Several studies have demonstrated that *Fn* can release OMVs, ranging in size between 30 and 250 nm. Despite limited comprehensive proteomic analyses, two studies have identified 98 and 991 proteins from *Fn* subsp. *animalis* 7/1 (EAVG_002) and *Fn* ATCC 23726, respectively. These include known and putative virulence [21,63]. According to these analyses, 72% and 40.98% of *Fn* OMVs are derived from the outer membrane, while proteins from the cytoplasm and periplasm are present in smaller proportions [21,63]. Most proteins found in *Fn* OMVs are associated with molecular binding and catalytic activities, displaying a broad functional range primarily involved in metabolic pathways [21]. Gas chromatography-mass spectrometry (GC-MS) analysis has revealed that LPS can constitute 60–70% mol/mol of *Fn* ATCC 51191-derived OMVs [69]. Thus, several proteins and molecules in *Fn* OMVs may exert biological effects on host cells. Therefore, the proteins and molecules in *Fn* OMVs may have biological effects on host cells.

*Fn* OMVs have been shown to be internalized by mouse bone marrow-derived macrophages, where they induce oxidative stress and promote the production of pro-inflammatory cytokines [115]. In contrast, *Fn* OMVs were not internalized in mouse gingival fibroblasts, but they did trigger apoptosis, an effect that was further amplified when fibroblasts were co-exposed to macrophages and *Fn* OMVs [115]. This suggests that Fn OMVs may interact with membrane receptors on gingival fibroblasts to mediate these apoptotic effects. Supporting this, *Fn* OMVs have been shown to exacerbate experimental periodontitis in vivo [115], indicating that similar pathogenic mechanisms may operate in a physiological setting.

Notably, *Fn* lacks the encoding for toxins, and few virulence factors have been proposed, with the primary focus in the literature on the adhesins FadA, Fap2, and FomA (Table 1) [116]. Consequently, we will explore the role of these proteins in the oral–gut axis and their potential implications in OMV-induced effects.

#### 4.2.1. FadA

The fusobacterial adhesin (FadA) plays a crucial role in bacterial attachment and invasion, implicating its involvement in carcinogenesis. FadA binds to an E-cadherin receptor, activating the β-catenin signaling pathway and oncogene expression in E-cadherin-expressing human colorectal cancer cell (CRC) lines and the non-CRC cell line HEK293 [117]. Additionally, FadA contributes to carcinogenesis through E-cadherin and Wnt/β-catenin-dependent induction of annexin A1 in proliferating cancer cells, subsequently activating the oncogene cyclin D [118]. Patients with colon carcinomas and precancerous adenomas show elevated *fadA* gene expression levels in colon tissue and an increased abundance of *Fn* in stool samples [117,119]. Moreover, *Fn* and *fadA* gene presence is heightened in UC, particularly in severe cases [120], indicating a potential pathogenic role in this condition. Similarly, increased detection of *Fn* and *fadA* has been reported in patients with periodontitis and non-orthodontic gingivitis [121].

Despite the evidence linking FadA to pathogenic roles in both the oral cavity and intestinal tract, no published reports directly prove the involvement of FadA-containing vesicles in the pathogenesis of periodontitis, oral cancer, CRC, and IBD. Recent work demonstrates that FadA-containing vesicles may exacerbate rheumatoid arthritis in a murine model [62]. The pathogenic role of *Fn* in experimental arthritis was effectively prevented by pre-treating *Fn* with GW4869, significantly reducing OMV release and subsequently lowering arthritis scores and inflammation [62]. FadA-containing OMVs can translocate to the joints, triggering synovial macrophage activation of vesicle trafficking through the Rab5a GTPase and subsequent activation of the pro-inflammatory transcriptional factor YB-1 [62]. These pro-inflammatory effects depend on the presence of FadA on OMVs and specific targeting of macrophages, as free FadA does not aggravate experimental arthritis. This suggests that OMV delivery of FadA to the joints is required to promote arthritis [62]. These findings align with increased *Fn* presence in samples from rheumatoid arthritis patients, correlating with disease severity, elevated FadA-containing OMVs, and increased Rab5a-YB-1 expression [62]. Thus, further research is necessary to determine whether a similar mechanism operates in the pathogenesis of oral and intestinal diseases.

#### 4.2.2. FomA

The porin FomA represents 30% of the proteins expressed in the *Fn* outer membrane and acts as a TLR2 agonist, promoting IL-8 secretion in HEK cells while failing to activate immune cells from TLR2 knockout mice [122]. Additionally, FomA exhibits antigenic activity, promoting anti-FomA antibody secretion in a TLR2-dependent manner [122]. Several studies have proposed its potential application in vaccination against *Fn* [59,123]. The presence of FomA in *Fn* ATCC 23726-derived OMVs has been implicated in the induction of NF-κB responses in human intestinal epithelial cells (IECs) via dynamin-mediated endocytosis in a TLR2-dependent manner [68]. This suggests that FomA-containing OMVs may interact with the human intestinal epithelium and promote pro-inflammatory responses. Various innate immune cells, including macrophages and DCs, express TLR2 [124]. Therefore, further research is required to comprehend the role of FomA-containing vesicles in regulating immune cell function within the oral–gut axis.

FomA-containing *Fn* OMVs have been shown to facilitate bacterial colonization in CRC. *Fn* OMVs fuse with the membrane of CRC cells DLD-1, a process inhibited by the endocytosis inhibitor fillipin but not by the dynamin-dependent inhibitor dynasore [25]. The FomA protein remains on the cell surface upon membrane fusion, enhancing Fn adhesion and autoaggregation by binding to the surface protein FN1441 (FruA). This adhesion was significantly reduced when using OMVs from a FomA-deficient *Fn* mutant strain [25]. FomA-containing *Fn* OMVs are also enriched in clinical CRC samples, further promoting *Fn* colonization in a colitis-associated CRC mouse model and accelerating cancer progression [25]. These findings underscore the critical role of FomA-containing OMVs in enhancing *Fn* distal colonization and contributing to CRC pathogenesis.

#### 4.2.3. Fap2

The lectin fibroblast activation protein 2 (Fap2) can bind the host-expressed polysaccharide D-galactose-β(1-3)-N-acetyl-D-galactosamine (Gal-GalNac), which is overexpressed in CRC. Intravenously infected Fap2-expressing *Fn* localizes to mouse tumor tissues, potentially correlating with the enrichment of *Fn* in CRC [64]. The reduction in Gal-GalNac levels in CRC tissue samples using O-glycanase resulted in diminished *Fn* binding [64]. Similarly, increased colonization of *Fn* in breast cancer occurs in a Fap-2-dependent manner via Gal-GalNAc binding [125].

Furthermore, Fap2 can inhibit antitumor immune responses in natural killer (NK) and T cells by binding to the inhibitory receptor TIGIT, potentially facilitating cancer cell evasion of immune responses [67]. Proteomic analysis has identified the presence of the autoregulated transporters D5REI9, D5RD69, and D5RBW2, which exhibit homology to Fap2 in *Fn*-derived OMVs [21]. However, another study using *Fn* OMVs did not detect Fap2 [62]. Therefore, the role of Fap2 in OMVs requires further elucidation.

#### 4.2.4. LPS

Another major constituent of *Fn* OMVs is LPS. *Fn*-derived LPS is immunogenic, eliciting serum IgM and IgG antibody responses in patients with periodontitis, indicating the induction of systemic immune responses against *Fn* [126]. The O-antigen of *Fn* strain 10953 LPS contains sialic acid, which binds to sialic acid-binding immunoglobulin-like lectins (Siglec) expressed on host innate immune cells [71].

A recent study demonstrated that LPS from *Fn* subsp. *animalis* (ATCC 51191) and *Fn*-derived OMVs bind to Siglec7 on human myeloid cells. Treatment with *Fn*-derived LPS or OMVs induced a pro-inflammatory profile in human monocyte-derived DCs, increasing TNF-α production. In contrast, human monocyte-derived macrophages exposed to *Fn*-derived LPS or OMVs exhibited increased IL-10 production, resembling an anti-inflammatory M2 phenotype, which could contribute to cancer progression [69]. Although Siglec 7 was shown to be necessary for inducing these effects on monocyte-derived dendritic cells and a human monocyte cell line, treatment with bacterial sialidase only caused a minor reduction in *Fn* binding to Siglec 7. This suggests that other LPS glycans in the O-antigen are required for Siglec 7 binding [69].

Traditionally, LPS from Gram-negative bacteria has been described as an activator of TLR4 signaling, leading to the subsequent induction of pro-inflammatory responses [127]. In accordance with this, *Fn* OMV-induced tumor growth and metastasis were shown to be mediated by TLR4 activation in vitro and in vivo. This suggests that OMV-derived LPS may promote cancer cell proliferation and migration. However, incubation of human intestinal epithelial cells (IECs) with *Fn*-derived OMVs induced only modest TLR4-dependent activation of IECs. This implies that LPS in *Fn* OMVs is not the primary contributor to the observed pro-inflammatory effects in IECs. Instead, these effects are more likely mediated by FomA-dependent TLR2 activation, as explained above [68]. Similarly, LPS neutralization using polymyxin B did not prevent *Fn* OMV-stimulated inflammation in macrophages and was attributed to their protein content [62]. Therefore, while LPS from *Fn* on OMVs may induce some pro-inflammatory effects on host cells, it is unlikely to be the sole factor responsible for the reported pro-inflammatory effects in the presence of *Fn* OMVs.

#### 4.2.5. *Fn* Non-Coding RNAs

Non-coding RNAs derived from *Fn* have been linked to colorectal cancer metastasis [128], and *Fn* has also been shown to induce transcriptional changes in both coding and non-coding RNA in THP-1-derived macrophages [129]. However, no studies currently associate non-coding RNAs contained within *Fn* OMVs with the regulation of host gene expression.

### 4.3. Effects of Fusobacterium nucleatum Outer Membrane Vesicles on Non-Immune and Immune Cells

Various studies have demonstrated that OMVs derived from *Fn* exhibit immunomodulatory properties in both epithelial and innate cells, with outcomes differing depending on the cell type. OMVs from *Fn* ATCC 23726 did not alter barrier permeability or affect tight junction protein expression in human intestinal epithelial cells. However, they induced the expression of pro-inflammatory NF-κB responses and stimulated IL-8 secretion in T84 cells while not impacting other intestinal epithelial cell lines such as HT-29, Caco-2, or HCTT16 [68]. A similar study reported that *Fn* OMVs and molecules above 50 kDa from *Fn* subsp. *polymorphum* ATCC 10953 induced NF-κB and MAPK responses, along with the secretion of IL-8 and TNF-α in a TLR4-dependent manner in the cancer cell line HT29 [108]. Comparable effects were observed using molecules above 50 kDa from *Fn* ATCC 10953 in human colonoid monolayers, indicating that these effects are not exclusive to cancer cell lines [108]. Notably, the latter experiments did not involve OMVs, necessitating further research to comprehend the interaction between *Fn*-derived OMVs and non-cancerous epithelial cells.

Nevertheless, the interaction between innate immune cells and epithelial cells may elicit distinct responses in the intestinal barrier in vitro and in vivo in the presence of *Fn* OMVs. Co-culture experiments involving human peripheral blood mononuclear cell (PBMC)-derived macrophages and Caco-2 cells revealed that *Fn*-derived OMVs could induce damage to the epithelial barrier through the activation of receptor-interacting protein kinase 1 (RIPK1)-mediated apoptosis, triggered by macrophage-derived TNF-α [130]. Additionally, *Fn*-derived OMVs induced macrophage polarization toward an M1-like pro-inflammatory phenotype, characterized by increased expression of CD86, elevated production of inducible nitric oxide synthase (iNOS) and reactive oxygen species (ROS), and reduced secretion of IL-10. Consequently, these OMVs may contribute to creating a pro-inflammatory environment in the intestinal barrier [130]. These findings were further supported in vivo, where intragastric administration of OMVs exacerbated DSS-induced colitis in mice, leading to increased gut barrier damage, elevated pro-inflammatory cytokine expression, and RIPK1-mediated apoptosis [130]. Consistent with this, the adoptive transfer of peritoneal macrophages treated with *Fn*-derived OMVs or induced toward an M1-like phenotype increased susceptibility to DSS-induced colitis, enhanced intestinal epithelial cell death, and reduced survival [130]. Similarly, another study reported that OMVs from *Fn* ATCC 25586 could induce pyroptosis, apoptosis, and necroptosis in the murine macrophage cell line RAW 264.7. These effects aligned with similar outcomes observed in vivo, where *Fn*-induced cell death mediated by Z-DNA-binding protein 1 (ZBP1) occurred in a model of apical periodontitis [131]. Thus, the interaction between *Fn*-derived OMVs and macrophages may trigger inflammation in both oral and intestinal barriers.

### 4.4. Effects of Fusobacterium nucleatum Outer Membrane Vesicles in Cancer

The tumor microenvironment may influence the composition of *Fn* OMVs, as evidenced by a recent study demonstrating that OMVs isolated from *Fn* ATCC 23726 cultured in an acidic environment resembling the tumor microenvironment exhibited approximately a 70% alteration in protein expression. This included the upregulation of several virulence factors, such as membrane occupation and recognition nexus (MORN) domain-containing proteins and type 5a secreted autotransporter (T5aSSs) [21]. Consistent with these findings, *Fn* ATCC 25586-derived OMVs were shown to promote lung tumor metastasis in a tumor-bearing mice model by inducing cancer-related autophagy. Pharmacological inhibition of autophagy using chloroquine effectively prevented OMV-induced metastasis [132]. Moreover, *Fn* ATCC 25586-derived OMVs enhanced the proliferation, migration, and invasion of breast cancer cell lines in vitro, and they contributed to increased tumor growth and liver metastasis in a murine model [133]. However, which OMV component(s) are responsible for this effect and whether *Fn* may participate in colorectal or oral cancer metastasis is unclear.

While much of the evidence suggests that *Fn* contributes to tumor progression, *Fn*-derived OMVs have been shown to induce PANoptosis, a form of inflammatory programmed cell death, in tumor cells by upregulating the executioner proteins gasdermin D/E (GSDMD/E) and mixed lineage kinase domain-like protein (MLKL), both of which inhibit ubiquitination [134]. Notably, the induction of PANoptosis by *Fn* OMVs, combined with the oncolytic virus HSV-1, has been proposed to enhance host antitumor immunity [134]. Further studies are needed to elucidate the specific *Fn* OMV components responsible for these antitumoral effects and to assess their potential application in cancer therapy.

## 5. *Akkermansia muciniphila*

### 5.1. Akkermansia muciniphila Interaction with Oral Pathobionts

Numerous reports suggest that the gut commensal *Akkermansia muciniphila* (*Am*) can mitigate experimental periodontitis and diminish the pathogenic potential of *Pg* and *Fn* [135,136,137]. In vitro, supplementation of *Am* to bone marrow macrophages infected with *Pg* induces an upregulation of IL-10 expression and a concurrent decrease in the pro-inflammatory cytokines IL-12 and TNF-α expression. Meanwhile, co-cultivation of both bacteria leads to reduced transcription of gingipain in *Pg* and an increased expression of Amuc_1100 and genes associated with the synthesis of monobactam-related antibiotics in *Am* [135,138]. Correspondingly, oral administration of *Am* or Amuc_1100 alleviates *Pg*-induced experimental periodontitis in a murine model, resulting in diminished tissue damage, elevated production of IL-10 in gingival tissue, and an increased frequency of anti-inflammatory M2 macrophages [136,138]. Furthermore, *Am* can enhance adhesion and promote the expression of tight junction proteins in gingival epithelial cells in vitro, potentially contributing to its beneficial effects during experimental periodontitis [135].

Recent evidence also indicates that *Am* can modulate *Fn* gene expression and inhibit *Fn*-induced periodontitis in mice. *Am* inhibits *Fn* growth and diminishes the expression of several virulence factors produced by *Fn*, such as FadA, Fap2, Aid1, FomA, and CmpA, in co-culture [137]. Additionally, transcriptomic analysis of gingival epithelial cells treated with both *Am* and *Fn* reveals that *Am* hinders *Fn*-induced inflammation by downregulating the TLR/MyD88/NF-κB signaling pathway and the expression of pro-inflammatory cytokines IL-1β, IL-6, and IL-8 [137]. In line with these findings, *Am* reduces *Fn*-induced experimental periodontitis, mitigating bone loss and decreasing the expression of the pro-inflammatory cytokines IL-1β and IL-6 in mouse periodontal tissue [137] (Figure 2).

### 5.2. Akkermansia muciniphila Outer Membrane Vesicles in the Gut Barrier

In contrast to the potentially pathogenic role of OMVs derived from oral pathobionts, OMVs from *Am* have been associated with beneficial effects in various disease models. Indeed, higher quantities of *Am* OMVs have been identified in fecal samples from healthy donors compared to patients with type 2 diabetes [139]. Additionally, orally administered *Am* OMVs can translocate into the large intestine and spread systemically into the liver, fat, muscle, and other organs 6 h post-administration [139]. Furthermore, treatment of Caco-2 cells with *Am* OMVs leads to increased expression of tight junction proteins, improving LPS-induced Caco-2 cells’ tight junction permeability by activating the tight junction regulator AMP-activated protein kinase (AMPK) [139,140]. Consistent with these findings, in vivo administration of *Am* OMVs protects mice from the increased gut barrier permeability observed in high-fat diet-fed mice [139].

*Am* OMVs can also directly modulate intestinal immune responses and microbiota composition. Direct delivery of *Am* OMVs into the intestinal lumen results in translocation to the Peyer’s patches and migration to the mesenteric lymph nodes, suggesting their ability to modulate immune responses [141]. Indeed, daily oral gavage of *Am* OMVs leads to increased DC activation in the Peyer’s patches, subsequent increased B cell activation, and augmented IgA production [141]. Interestingly, an elevated level of IgA-coated bacteria was detected in mice treated with *Am* OMVs, suggesting a potential role in reducing pathogenic bacteria and/or promoting commensal colonization in the gut microbiota [141]. Consistent with these findings, *Am* OMVs induce tolerogenic human DCs and increase IL-10 production in vitro by modulating microRNA expression related to a pro-inflammatory profile [142].

Another recent study demonstrated that daily oral gavage of *Am* OMVs to DSS-treated mice promotes the colonization of beneficial commensals, including various *Bacteroides* species. This was attributed to specific OMV membrane fusion with commensals, increasing bacterial proliferation [141]. Additionally, this treatment ameliorates DSS-induced colitis, improves gut barrier integrity by increasing mucin-producing goblet cells and tight junction proteins, and enhances the production of IL-10 and IL-13 in the colon [141].

### 5.3. Akkermansia muciniphila Outer Membrane Vesicles in Cancer

The administration of the *Am* OMVs can enhance the efficacy of cancer immunotherapy, particularly when combined with anti-PD-1 treatment, as demonstrated in a murine model of colorectal cancer [141]. Additionally, *Am* OMVs exhibit the capability to induce antitumor immunity in a murine model of prostate cancer. This is evidenced by increased granzyme- and IFN-γ-expressing CD8^+^ T cells and heightened recruitment of M1-like macrophages [143]. These findings suggest that *Am* OMVs may not only ameliorate dysbiosis induced by oral pathobionts in the gut but also prevent gut inflammation and colorectal cancer mediated by *Fn* (Figure 2). The specific *Am*-derived proteins responsible for these anti-inflammatory and anti-tumoral effects on host cells are currently unknown, and whether the Amuc_1100 protein is present on OMVs remains to be elucidated. Further research is essential to explore the potential beneficial effects of *Am* OMVs in the oral barrier, with potential implications for promoting the expansion of beneficial commensals, preventing the proliferation of oral pathobionts, and modulating antimicrobial responses in the oral cavity.

## 6. Future Directions and Conclusions

Oral pathobionts can contribute to disease pathogenesis in the gut and other distal organs through direct colonization of distal tissues, distal alterations of the tissue-resident microbiota, and/or the release of OMVs containing virulence factors and pro-inflammatory molecules. The prevailing literature reviewed herein suggests that proteins associated with virulence can be released within pathobiont-derived OMVs, activating inflammatory responses in non-immune and immune cells and thereby contributing to local and distal disease pathogenesis.

The effects of *Pg* OMVs on host cells and systemic dissemination are diverse and significant. *Pg*-derived OMVs not only demonstrate the ability to promote immune evasion by inducing tolerance in immune cells, highlighting the context-dependent role of OMVs, but also exhibit immunostimulatory effects by inducing the maturation of dendritic cells, promoting the differentiation of T cells, and increasing the expression of inflammatory cytokines in macrophages. On the other hand, they also exert immunoregulatory effects by inducing monocyte tolerance, compromising neutrophil function, and limiting the pro-inflammatory response in macrophages. Environmental factors such as pH and nutrient availability can modulate the protein content of OMVs, providing the pathobiont with a potential advantage in regulating host responses.

Furthermore, *Pg* OMVs impact non-immune cells by inducing the expression of inflammatory mediators, altering cell proliferation and migration, and affecting endothelial permeability. Their systemic effects include the traversal of barriers to reach the hippocampus and cortex, where they induce neuroinflammation and contribute to cognitive impairment. In diabetic retinopathy, *Pg* OMVs were found to increase the permeability of the blood–retinal barrier and exacerbate microvasculature alterations.

Thus, these findings suggest that *Pg* OMVs play a significant role in modulating the host immune response, cellular functions, and systemic physiology, highlighting their potential implications in various diseases and health conditions. Further research is crucial for a comprehensive understanding of the mechanisms underlying these effects and their implications in specific pathological conditions.

The research on the effects of *Fn* OMVs has provided compelling evidence of their immunomodulatory properties in both innate and epithelial cells. It has been demonstrated that these OMVs can induce pro-inflammatory responses, influence epithelial barrier integrity, and promote cancer cell proliferation and metastasis. Furthermore, the interaction between *Fn* OMVs and macrophages has been shown to trigger inflammation in oral and intestinal barriers, exacerbating conditions such as colitis. The distinct responses of *Fn* OMVs in different cell types suggest their multifaceted role in modulating the host microenvironment. However, further research is warranted to elucidate the specific components and mechanisms underlying these effects and their potential implications in different types of cancer metastasis. Understanding the interplay between *Fn* OMVs and host cells is crucial for developing targeted therapeutic strategies to mitigate their impact on host physiology and disease progression.

The various studies and evidence presented strongly support the role of *Akkermansia muciniphila* in mitigating periodontitis and its potential beneficial effects in various disease models. The ability of *Am* to modulate immune responses, inhibit the growth and virulence of pathogenic bacteria, and promote the expansion of beneficial commensals highlights its potential as a therapeutic target for various diseases, including periodontitis, gut barrier dysfunction, and cancer.

The findings on *Am* OMVs reveal their ability to enhance gut barrier integrity, modulate immune responses, and even improve the efficacy of cancer immunotherapy. These results open up new possibilities for using *Am* OMVs to prevent dysbiosis, inflammation, and cancer progression. However, further research is necessary to fully understand the mechanisms and potential application of *Am* OMVs in the oral barrier. Investigating the presence of specific *Am*-derived proteins on OMVs and their effects on oral microbiota and antimicrobial responses could pave the way for novel therapeutic approaches in oral health.

The biotechnological implications of these studies revolve around the potential uses of OMVs from oral pathobionts in creating new treatment approaches and preventing diseases associated with these pathobionts. Some key prospects include: (1) Designing methods to prevent and treat periodontitis and systemic conditions linked to oral pathobionts, such as rheumatoid arthritis, Alzheimer’s disease, type 1 diabetes, cardiovascular diseases, diabetic retinopathy, and certain cancers. (2) Harnessing the antimicrobial properties of OMVs from *Am* to regulate the interaction between pathobionts and the host, potentially averting disease development in the oral–gut axis. (3) Exploring whether OMVs from oral pathobionts can encourage favorable commensal growth and enhance tissue barrier integrity while triggering anti-inflammatory effects in the gastrointestinal tract and distant tissues. (4) Investigate whether using OMVs from oral pathobionts as biomarkers could aid early disease detection or serve as potential therapeutic targets. (5) Developing innovative diagnostic tools for monitoring the transmission of oral pathogens’ impact on host health. Further exploration is required into how these OMVs interact with host cells and investigating applications of *Am*-derived OMV’s role in shaping interactions among various components within the body systems, including prevention against the onset of ailment manifestation at sites like the oral–gut interface.

All the evidence underscores the crucial role of OMVs in fomenting local inflammation in the oral cavity and in the pathogenesis of diseases in the gastrointestinal tract and beyond.

## Figures and Tables

**Figure 1 ijms-25-11141-f001:**
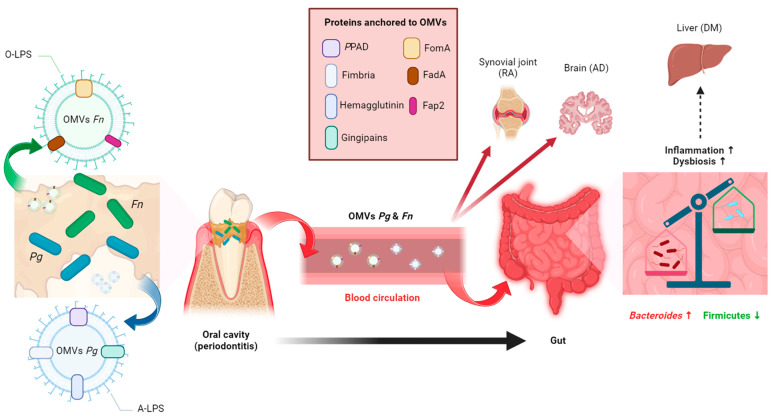
Inflammation and gut dysbiosis induced by oral pathobionts. Outer membrane vesicles (OMVs) of *Porphyromonas gingivalis* (*Pg*) and *Fusobacterium nucleatum* (*Fn*) transport various virulence factors from the oral cavity to the gut and other distal organs via blood circulation. Subsequently, *Pg* and *Fn* could induce inflammation and gut dysbiosis through their OMVs. Created with BioRender (https://www.biorender.com/, accessed on 1 June 2024).

**Figure 2 ijms-25-11141-f002:**
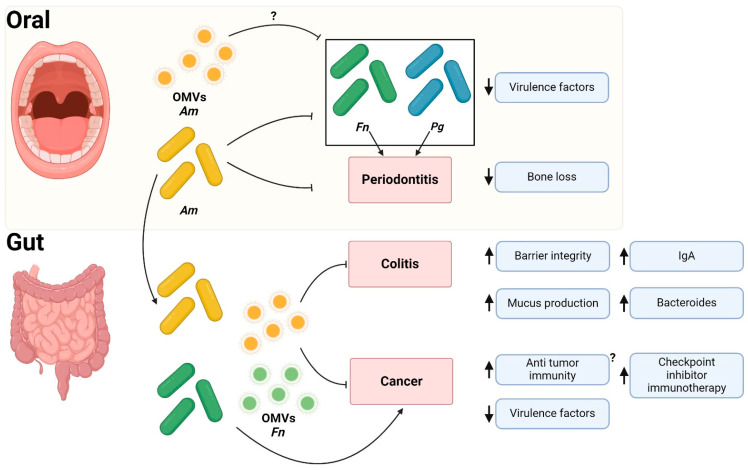
Proposed model for *Akkermansia muciniphila*-mediated regulation of oral pathobiont-induced disease at the oral–gut axis. *Akkermansia muciniphila* (*Am*) and its OMVs are suggested to exert protective effects, mitigating periodontitis in experimental models and reducing the expression of virulence factors in *Porphyromonas gingivalis* (*Pg*) and *Fusobacterium nucleatum* (*Fn*). In the gut, *Am* contributes to maintaining barrier integrity and promoting the expansion of beneficial commensal species, thereby reducing the severity of colitis in mice. Additionally, *Am* may potentiate the efficacy of cancer immunotherapy. Up and down arrows indicate an increase or decrease in each indicated parameter, respectively. Question marks (shown as “?”) indicate areas where evidence is still inconclusive or remains to be explored. Created with BioRender.

**Table 1 ijms-25-11141-t001:** Pathobiont-expressed virulence factors associated with OMVs.

Pathobiont	Virulence Factor	Function	Cell Target/Molecule	Pathology	References
*P. gingivalis*	A-LPS	Production of inflammatory mediators through TLRs, causing systemic inflammation in the host.	DC, Mo, MØ, PMN (neutrophil)	PD, AD, RA	[48,49]
	PPAD	Citrullination of peptides confers evasion of the host immune system and is involved in the biogenesis of OMVs.	Peptides with Arg residues, preferably at the C-terminus	PD, RA	[22,50,51,52,53,54]
	*Gingipains* (Rgps, Kgp)	Cleavage of specific residues (RgpA/B in Arg and Kgp in Lys) of peptides and confers the ability to evade the host’s immune system. Degradation of host adhesins and tight junction proteins.	Hepatocytes, brain microvascular endothelial cells, vascular endothelial cells	PD, AD, CVD, DM	[36,55,56,57]
	Fimbria (FimA, C, D and E, Mfa1)	Group of proteins involved in adhesion, invasion of host cells, and mediation of vesiculation of OMVs.	Keratinocytes, fibroblasts, DC	PD	[39]
	Hemagglutinin and hemolysin	Agglutination (hemagglutinin)/lysis (hemolysin) of erythrocytes to obtain nutrients (heme) in the host.	Erythrocytes	PD	[48]
	sRNA45033	Induction of apoptosis and production of inflammatory mediators in the host.	hPDLC	PD	[58]
*F. nucleatum*	FomA	Porin that facilitates host cell invasion. Promotes pro-inflammatory response by binding TLR2. Promotion of binding to CRC cells and *Fn* autoaggregation.	EC, IEC	PD, CRC	[25,59,60,61]
	FadA	Adhesion protein that facilitates binding to host cells. Binds to E-cadherin receptor, activating Wnt-β-catenin pathway, promotes macrophage activation in synovial tissue through Rab5-YB-1 pathway	EC, IEC, moMØ, MØ	PD, CRC, AR, UC	[61,62,63]
	Fap2	Adhesion protein that facilitates binding to host cells. Binds to Gal-GalNAc and TIGIT inhibitory receptor, affecting NK response	IEC, DC, TL, NK	PD, CRC	[64,65,66,67,68]
	MORN-domain containing proteins	Protein that enhances bacterial adherence and active invasion, probably through autotransporter genes upregulation.	Not fully characterized		[21,63]
	O-LPS	Production of inflammation mediators through binding to Siglec 7 and activation of TLR4 pathway	DC, moMØ, MØ, moDC	PD, CRC	[69,70,71]

AD: Alzheimer disease; CRC: colorectal cancer; CVD: cardiovascular disease; DC: dendritic cells; DM: diabetes mellitus; EC: epithelial cell; IEC: intestinal epithelial cell; moDC: monocyte-derived dendritic cells; moMØ: monocyte-derived macrophages; MØ: macrophage; NK: natural killer; PD: periodontitis; PMN: polymorphonuclear; RA: rheumatoid arthritis; TL: T lymphocyte; UC: ulcerative colitis.

## Data Availability

No new data were created or analyzed in this study. Data sharing does not apply to this article.

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
