# Peer review of "Oral Pathobiont-Derived Outer Membrane Vesicles in the Oral–Gut Axis"

_ijms, 2024, doi:10.3390/ijms252011141_

Round 1
Reviewer 1 Report
Comments and Suggestions for Authors
This is a fairly well-written and comprehensive review. There are no major concerns. However, the authors may consider including two additional studies that suggest the pootential role of vesicles in platelet aggregation and modulation of bacterial invasion. The references for these studies are as follows:
Porphyromonas gingivalis vesicles enhance attachment, and the leucine-rich repeat BspA protein is required for invasion of epithelial cells by "Tannerella forsythia".
Inagaki S, Onishi S, Kuramitsu HK, Sharma A. Infect Immun. 2006 Sep;74(9):5023-8. doi: 10.1128/IAI.00062-06.PMID: 16926393
Porphyromonas gingivalis platelet aggregation activity: outer membrane vesicles are potent activators of murine platelets
A Sharma 1, E K Novak, H T Sojar, R T Swank, H K Kuramitsu, R J Genco
PMID: 11154438
Author Response
|
1. Summary |
|
|
Thank you very much for taking the time to review this manuscript. Please find the detailed responses below and the corresponding revisions/corrections highlighted in yellow in the re-submitted files. |
|
|
2. Point-by-point response to Comments and Suggestions for Authors |
|
|
Comments 1: This is a fairly well-written and comprehensive review. There are no major concerns. However, the authors may consider including two additional studies that suggest the potential role of vesicles in platelet aggregation and modulation of bacterial invasion. The references for these studies are as follows: Porphyromonas gingivalis vesicles enhance attachment, and the leucine-rich repeat BspA protein is required for invasion of epithelial cells by "Tannerella forsythia". Inagaki S, Onishi S, Kuramitsu HK, Sharma A. Infect Immun. 2006 Sep;74(9):5023-8. doi: 10.1128/IAI.00062-06.PMID: 16926393
Porphyromonas gingivalis platelet aggregation activity: outer membrane vesicles are potent activators of murine platelets A Sharma 1, E K Novak, H T Sojar, R T Swank, H K Kuramitsu, R J Genco PMID: 11154438 |
|
|
Response 1: We appreciate the reviewer's comments, and we have incorporated these suggested references into the main text in section 3.3, which discusses the general roles of Pg OMVs (lines 170-173), and section 3.7, which discusses Pg OMVs' contribution to systemic diseases (lines 476-482).
|
|

Reviewer 2 Report
Comments and Suggestions for Authors
My comments are in the attached. I suggest some modifications.

Author Response
|
1. Summary |
|
|
|
Thank you very much for taking the time to review this manuscript. Please find the detailed responses below and the corresponding revisions/corrections highlighted in yellow in the re-submitted files.
|
||
|
2. Point-by-point response to Comments and Suggestions for Authors |
||
|
Comment 1: The repeated flaw in the review is lack of mechanistic insight for each topic contributing to understanding where and to what degree OMV has importance in pathogenesis or therapy. |
||
|
Response 1: We thank the reviewer for raising this issue. Based on the reviewer's suggestions, we have added a more extensive analysis of the potential mechanisms involved in oral pathobiont-derived OMV pathogenesis and therapy. We have added a section discussing how oral pathobiont-derived OMVs can be internalized into host cells and discuss recently published mechanisms mediated by gingipain-containing Pg OMVs (section 3.3, lines 246-253) and FomA-containing Fn OMVs (section 4.2, lines 617-626). In addition, we have added a recent study discussing the antitumoral effects of Fn OMVs, promoting cancer cell PANoptosis (section 4.4, lines 736-744). Comment 2: There is a lack of characterization of OMV in context with host, exosomes, multilamellar vesicles, or oral pathogen responses to the molecular signals on display in these additional cell vehicles that trigger both microbial and host metabolic, molecular and immunologic functions. Response 2: We thank the reviewer for this comment. We believe that an extensive description of oral pathobiont-derived OMVs and how host-derived vesicles can interact with oral pathobionts is out of the scope of our manuscript, considering the lack of evidence in the literature. We found two articles describing that Pg can invade bone marrow-derived dendritic cells, increasing the secretion of host exosomes, which may contribute to altered immunity and periodontitis (PMID: 34141629, PMID: 36818565). However, these effects were independent of Pg OMVs and it is still unknown if Pg OMVs can trigger alterations in exosome secretion or that host exosomes can alter Pg virulence and/or OMV release. Comment 3: Furthermore, we note that bacterial plasmids released by the viral component of the microbiome will influence oral pathogen OMV responses and release. Response 3: We thank the reviewer for this comment. Although increasing evidence indicates that bacterial OMVs can interact with phages, which may control bacterial susceptibility to phage infection (recently reviewed in PMID: 39304539), we did not find any articles reporting these interactions in the context of oral pathobiont-derived OMVs. Therefore, we believe that discussing this topic is out of the scope of the manuscript. Comment 4: There is also no attempt to integrate into the discussion the array of host response receptor activities such as TIMS with only short notes of receptors sites related to mucosal cell integrity and attachment. Response 4: We thank the reviewer for this comment. We have added a paragraph discussing the contribution of various host receptors to oral pathobiont-derived OMV in the context of Pg OMVs (lines 428-442). However, we have focused on the available literature related to Pg- and Fn-derived OMVs, which has mainly reported interactions with membrane bound-pattern recognition receptors such as TLRs and we can only suggest the involvement of other receptors based on evidence from virulence factors and their targets on host cells. Of note, TIM receptors such as TIM-4 mainly bind phosphatidylserine, which is not commonly found on the membrane of Pg or Fusobacteria. We have added a short paragraph addressing this topic (lines 428-442). Comment 5: Moreover, increasingly we recognize the release from oral pathogens a variety of metallo-endopeptidases and isomerases of various types mediating the gingipain inflammatory activities in host cells. Response 5: We thank the reviewer for this comment, but we are unsure how to address this request. We did not find any works linking oral pathobiont-derived OMVs with bacterial metalloproteinases and isomerases or about the interaction between oral-pathobiont OMVs and host metalloproteinases. We believe that it is out of the scope of our manuscript to delve deeper into this topic. However, two older articles describe the potential role of Pg endopeptidase O (PepO) in host cell invasion and intracellular survival (PMID: 14622347; PMID: 15213115). Proteomic analysis identified PepO within membrane and cytosolic fractions of Pg as well as in one of two preparations of Pg OMVs (PMID: 24620993). Therefore, we have included the following paragraph (lines 183-189): Studies indicate that Pg metallo-endopeptidase PepO, a homologous to human endothelin-converting enzyme (ECE)-1, may play a key role in invasion and intracellular survival of Pg in human gingival epithelial cells (PMID: 14622347; PMID: 15213115). PepO is localized on the Pg membrane and has also been found in Pg OMVs, suggesting its potential contribution to the entry of Pg OMVs into host cells. Comment 6: There is also an array of host membrane proteins that organize signal responses that are identifiable both at plasma membrane and endosomal-lysosomal membrane that require discussion as they relate to endocytosis of both Pg and Fn and ultimately host response to OMV. Response 6: We thank the reviewer for highlighting this important topic. We have expanded section 2 introducing the mechanisms by which oral pathobiont-derived OMVs can interact with host cells and/or be internalized (lines 96-108), in the context of Pg OMVs (section 3.3, lines 174-182; section 3.4, lines 382-387) and Fn OMVs (section 4.2, lines 559-567). We would like to point out that we have discussed these topics in the context of Fn and Pg OMVs and we believe that discussing the mechanisms described for interactions between host cells and other bacterial OMVs is out of the scope of our manuscript. Comment 7: In addition, metabolic bacteria activity as noted can have a broad effect with both essential and nonessential amino acid synthesis pathways and their secondary downstream expressions further characterizing any long-term significance to OMV release which may be more effective during acute interaction compared to longer term, chronic pathology. Response 7: We thank the reviewer for raising this question. We have added to the Pg section, an article that discusses the effect of Pg metabolic activity on host liver amino acid synthesis (section 3.3, lines 266-273). However, we did not find any articles to date reporting the effect of oral pathobiont-derived OMVs on host amino acid metabolism. Comment 8: Lastly there is an additional need to place the structural element of OMV in context with non-structural molecule activities that are also highly contributory to any clinical manifestation driven by the OMV (e.g., effect on genomic expressions and alteration to genome). Response 8: We thank the reviewer for this comment. We have added a paragraph discussing the role of non-coding RNA present in Pg OMVs and their effects on the host cell genome (section 3.3, lines 356-367). We did not find any reports describing non-coding RNA in Fn OMVs but we added a short sentence addressing this gap in the literature (section 4.2, lines 674-679). |
||

Reviewer 3 Report
Comments and Suggestions for Authors
Dear authors
The review is very complete, we know that it is not possible to include all the scientific evidence that can be found about the research topic in a document. However, you mention rheumatoid arthritis associated with P. gingivalis among systemic diseases.
I recommend the item
3.3. Porphyromonas gingivalis outer membrane vesicles
Gingipains and Peptidylarginine deiminase (PPAD): I suggest including findings that may be important in your review in which you associate anti-RgpA antibodies with arthritis, where the antigen is obtained from membrane vesicles, which is an important part of your review. I consider that you can strengthen this association:
Castillo, D. M., Lafaurie, G. I., Romero-Sánchez, C., Delgadillo, N. A., Castillo, Y., Bautista-Molano, W., Pacheco-Tena, C., Bello-Gualtero, J. M., Chalem-Choueka, P., & Castellanos, J. E. (2023). The Interaction Effect of Anti-RgpA and Anti-PPAD Antibody Titers: An Indicator for Rheumatoid Arthritis Diagnosis. Journal of clinical medicine, 12(8), 3027. https://doi.org/10.3390/jcm12083027
I know that the vast majority of information on F. nucleatum is associated with cancer, however, other systemic repercussions should be taken into account, with recent information such as that reported by:
Zhou, L. J., Lin, W. Z., Meng, X. Q., Zhu, H., Liu, T., Du, L. J., Bai, X. B., Chen, B. Y., Liu, Y., Xu, Y., Xie, Y., Shu, R., Chen, F. M., Zhu, Y. Q., & Duan, S. Z. (2023). Periodontitis exacerbates atherosclerosis through Fusobacterium nucleatum-promoted hepatic glycolysis and lipogenesis. Cardiovascular research, 119(8), 1706–1717. https://doi.org/10.1093/cvr/cvad045
It would be wonderful if the authors could make a graphical proposal of the action of Akkermansia muciniphila against the virulence factors of P. gingivalis, F. nucleatum, and their OMVs, as well as how the findings of the literature reviewed and included in this document can prevent or affect the pathogenesis of these bacteria in the systemic diseases for which there is evidence. It would be something like what they have in the conclusions but graphically.
Indicate to the editors that I require major corrections to confirm the new version, but not because they are major corrections.
Cordially,
Author Response
|
1. Summary |
|
|
Thank you very much for taking the time to review this manuscript. Please find the detailed responses below and the corresponding revisions/corrections highlighted in yellow in the re-submitted files. |
|
|
2. Point-by-point response to Comments and Suggestions for Authors |
|
|
Comments 1: Dear authors The review is very complete, we know that it is not possible to include all the scientific evidence that can be found about the research topic in a document. However, you mention rheumatoid arthritis associated with P. gingivalis among systemic diseases. I recommend the item 3.3. Porphyromonas gingivalis outer membrane vesicles Gingipains and Peptidylarginine deiminase (PPAD): I suggest including findings that may be important in your review in which you associate anti-RgpA antibodies with arthritis, where the antigen is obtained from membrane vesicles, which is an important part of your review. I consider that you can strengthen this association: Castillo, D. M., Lafaurie, G. I., Romero-Sánchez, C., Delgadillo, N. A., Castillo, Y., Bautista-Molano, W., Pacheco-Tena, C., Bello-Gualtero, J. M., Chalem-Choueka, P., & Castellanos, J. E. (2023). The Interaction Effect of Anti-RgpA and Anti-PPAD Antibody Titers: An Indicator for Rheumatoid Arthritis Diagnosis. Journal of clinical medicine, 12(8), 3027. https://doi.org/10.3390/jcm12083027 |
|
|
Response 1: We thank the reviewer for his/her comments and the positive evaluation of our manuscript. We have incorporated the suggested reference into the main text (section 3.3, lines 342-348). Comment 2: I know that the vast majority of information on F. nucleatum is associated with cancer, however, other systemic repercussions should be taken into account, with recent information such as that reported by: Zhou, L. J., Lin, W. Z., Meng, X. Q., Zhu, H., Liu, T., Du, L. J., Bai, X. B., Chen, B. Y., Liu, Y., Xu, Y., Xie, Y., Shu, R., Chen, F. M., Zhu, Y. Q., & Duan, S. Z. (2023). Periodontitis exacerbates atherosclerosis through Fusobacterium nucleatum-promoted hepatic glycolysis and lipogenesis. Cardiovascular research, 119(8), 1706–1717. https://doi.org/10.1093/cvr/cvad045 Response 2: We thank the reviewer for his/her comments, and we have incorporated this reference into section 4.1 (lines 533-541). Interestingly, we also found that Pg can induce metabolic alterations in the liver (section 3.3, lines 266-273), suggesting that both oral pathobionts can alter host metabolism and promote liver disease pathogenesis. Comment 3: It would be wonderful if the authors could make a graphical proposal of the action of Akkermansia muciniphila against the virulence factors of P. gingivalis, F. nucleatum, and their OMVs, as well as how the findings of the literature reviewed and included in this document can prevent or affect the pathogenesis of these bacteria in the systemic diseases for which there is evidence. It would be something like what they have in the conclusions but graphically. Response 3: As requested by the reviewer, we have incorporated a new figure proposing how Akkermansia muciniphila can interact and prevent oral pathobiont-induced pathogenesis (Figure 2). |
|
